# Single-cell transcriptome profiling of the vaginal wall in women with severe anterior vaginal prolapse

Yaqian Li [1,9], Qing-Yang Zhang [2,3,9], Bao-Fa Sun[2,4,5,6,9], Yidi Ma[7], Ye Zhang[7], Min Wang[8], Congcong Ma[7], Honghui Shi[7], Zhijing Sun[7], Juan Chen[7], Yun-Gui Yang [2,4,5,6 ✉] & Lan Zhu [7 ✉]

Anterior vaginal prolapse (AVP) is the most common form of pelvic organ prolapse (POP) and has deleterious effects on women's health. Despite recent advances in AVP diagnosis and treatment, a cell atlas of the vaginal wall in AVP has not been constructed. Here, we employ single-cell RNA-seq to construct a transcriptomic atlas of 81,026 individual cells in the vaginal wall from AVP and control samples and identify 11 cell types. We reveal aberrant gene expression in diverse cell types in AVP. Extracellular matrix (ECM) dysregulation and immune reactions involvement are identified in both non-immune and immune cell types. In addition, we find that several transcription factors associated with ECM and immune regulation are activated in AVP. Furthermore, we reveal dysregulated cell–cell communication patterns in AVP. Taken together, this work provides a valuable resource for deciphering the cellular heterogeneity and the molecular mechanisms underlying severe AVP.

[1] Medical Science Research Center, Peking Union Medical College Hospital, Chinese Academy of Medical Science and Peking Union Medical College, 100730 Beijing, China. [2] CAS Key Laboratory of Genomic and Precision Medicine, Collaborative Innovation Center of Genetics and Development, College of Future Technology, Beijing Institute of Genomics, Chinese Academy of Sciences, 100101 Beijing, China. [3] Sino-Danish College, University of Chinese Academy of Sciences, 101408 Beijing, China. [4] China National Center for Bioinformation, 100101 Beijing, China. [5] University of Chinese Academy of Sciences, 100049 Beijing, China. [6] Institute of Stem Cell and Regeneration, Chinese Academy of Sciences, 100101 Beijing, China. [7] Departments of Obstetrics and Gynecology, Peking Union Medical College Hospital, Chinese Academy of Medical Sciences and Peking Union Medical College, 100730 Beijing, China. [8] Department of Rheumatology and Clinical Immunology, Peking Union Medical College Hospital, Chinese Academy of Medical Sciences and Peking Union Medical College, 100730 Beijing, China. [9] These authors contributed equally: Yaqian Li, Qing-Yang Zhang, Bao-Fa Sun. ✉email: ygyang@big.ac.cn; zhulan@pumch.cn

Pelvic organ prolapse (POP) is a major health issue for women in which pelvic organs, such as the uterus, bladder and rectum, protrude from the vagina due to weakness of the supportive tissue, leading to bladder and bowel dysfunction[1,2]. Symptomatic POP significantly affects quality of life, causing discomfort, pain, and embarrassment[3]. The prevalence of symptomatic POP is 6–9.6% in women aged >20 years and 30–40% in postmenopausal women[3,4]. The lifetime risk of undergoing surgery for prolapse is 11−20%[5,6]. Despite the high incidence of POP, little is known about its pathophysiological course. Therefore, investigation of the pathophysiological changes and molecular mechanism in POP is urgently needed.

Anterior vaginal prolapse (AVP) is the most frequently occurring form of POP. Recently, some studies have reported that the thickness, mechanical properties and structural composition of the vaginal wall are altered in women with POP[7,8]. Weakness of the vaginal wall is considered to be a possible cause of prolapse. Histological and biochemical alterations in the vaginal wall have been widely studied, revealing that the main constituents in vaginal connective tissues—collagen and elastin fibers—are altered and that an imbalance in matrix metalloproteinases (MMPs) and tissue inhibitors of metalloproteinase (TIMPs) leads to dysregulation of extracellular matrix (ECM) metabolism, which in turn influences the architectural remodeling of the vaginal muscularis propria[9]. However, these findings have been identified mainly by immunohistochemistry and western blotting and only partially represent the changes in the vaginal wall in POP. Some researchers have investigated genome and transcriptome alterations in the prolapsed vaginal wall[10–13]. Their findings indicated that changes in the immune response, estrogen related receptor expression, signaling pathways or other candidate genes (*LAMC1*, *LOXL-1*, and *Fibulin-5*) potentially mediate the development of POP[14–19], implying explanations for the pathogenesis of POP at the gene level[10,20]. Although previous studies have documented histological alterations and potential critical genes, this evidence is not strong enough to confirm and decipher the genetic and molecular mechanisms of POP. Thus, more comprehensive and in-depth investigations of specific and explicit molecular mechanisms need to be conducted to provide useful insights into POP pathogenesis.

The cellular composition of the vaginal wall is complex, comprising primarily fibroblasts and smooth muscle cells (SMCs), which play important roles in the extracellular integrity and mechanical stretching of the anterior vaginal wall[21,22]. However, the complete cell type composition and aspects of cellular heterogeneity leading to vaginal prolapse remain largely unknown. Vaginal fibroblastic cells derived from prolapsed tissues display altered functional characteristics[23,24] in vitro and alterations in the expression of some genes compared with fibroblastic cells derived from non-prolapsed sites[24–26]. In addition, the muscle fibers appear disrupted and altered, suggesting that they eventually contribute to the dynamic function of the vaginal wall[27]. Although this study demonstrated that these two cell types in the vaginal wall may play critical roles in the etiology of prolapse, they focused only on a certain type of vaginal cell and ignored cellular heterogeneity. Moreover, to understand the molecular mechanism underlying the prolapse process, it is important to investigate the cellular composition and cell type-specific changes in gene expression in normal and prolapsed vaginal walls. Hence, systematic and in-depth studies on cell type-specific composition and function need to be carried out to evaluate the detailed molecular mechanisms underlying severe POP in order to comprehensively understand the etiology of prolapse.

The rapid development of single-cell RNA sequencing (scRNA-seq), which enables specific profiling of cell populations and gene expression at the single-cell level, allows us to investigate the composition of cells involved in the pathogenesis of AVP and elucidates cell type-specific molecular alterations at the single-cell level. Our study provides a comprehensive atlas of transcriptome data for vaginal cell types in normal and prolapsed human vaginas. Notably, we reveal alterations in gene expression at the level of cell type specificity and reveal the roles of vaginal cells in ECM remodeling and the immune response during prolapse. We demonstrate the changes in both non-immune and immune cells in the etiology of POP. Thus, our work provides a comprehensive understanding of the molecular mechanism of prolapse at the single-cell level and enhances the understanding of the pathophysiological process of severe AVP, which offers insights for improving current preventative and therapeutic strategies of this disorder.

## Results

**Single-cell transcriptome atlas and cell typing in POP and control samples.** To understand the cellular diversity and molecular features of the human vaginal wall in POP patients, we used standard methods to isolate vaginal wall cells from the prolapsed anterior vaginal wall of 16 AVP patients and the normal anterior vaginal wall of 5 control individuals undergoing hysterectomy and then performed scRNA-seq using 10× Chromium Genomics protocols (Fig. 1a). The overall pelvic organ prolapse quantification (POP-Q) stage of AVP was 3-4 (Fig. 1b and Table 1 and Supplementary Data 1). Histological and morphological changes were confirmed with haematoxylin and eosin (H&E) staining and α-SMA immunohistochemical (IHC) staining, which assessed alterations in smooth muscle (Fig. 1c and Supplementary Fig. 2a). As expected, the morphology of each layer in the POP samples was altered compared with that in the control samples, and the muscularis exhibited atrophy. The extensive changes in the prolapsed vagina at the tissue level prompted us to investigate the molecular mechanism underlying POP.

After an initial quality control step (see "Methods"), we obtained a total of 81,026 single cells from normal and prolapsed vaginal walls, in which the expression of a median of 1600 genes per cell could be detected (Supplementary Fig. 1a). Of these 81,026 single cells, 65,434 cells originated from prolapsed vaginal walls, and 15,592 cells originated from normal vaginal walls (Supplementary Fig. 1b). To define cell types, we first processed the 10× data using Seurat R packages for quality control, normalization and batch effect correction and then performed doublet removal with DoubletFinder (see "Methods"). Then, the principal component analysis was applied for dimensionality reduction. We further used unsupervised graph clustering to partition the cells into clusters and visualized the clusters via uniform manifold approximation and projection (UMAP) (Fig. 1d and Supplementary Fig. 1b). To determine the cellular identity of each cluster, we generated cluster-specific marker genes via differential gene expression analysis (Supplementary Data 2 and Supplementary Data 3). In most cases, well-known cell type markers, such as *LUM* for fibroblasts[28], *TAGLN* for smooth muscle cells[29], and *AIF1* for macrophages[30,31], were used to determine the cellular identity of the clusters (Fig. 1e, f and Supplementary Fig. 1c). In addition to the well-known markers, we also identified multiple additional markers, for example, *FBLN1* for fibroblasts and *MT1A* and *PLN* for smooth muscle cells (Supplementary Data 3). In total, 11 cell types were identified in POP and control samples based on the canonical markers: epithelial cells, fibroblasts, smooth muscle cells, myoepithelial cells, endothelial cells, lymphatic endothelial cells, macrophages, T cells, B cells, plasma B cells and mast cells. The

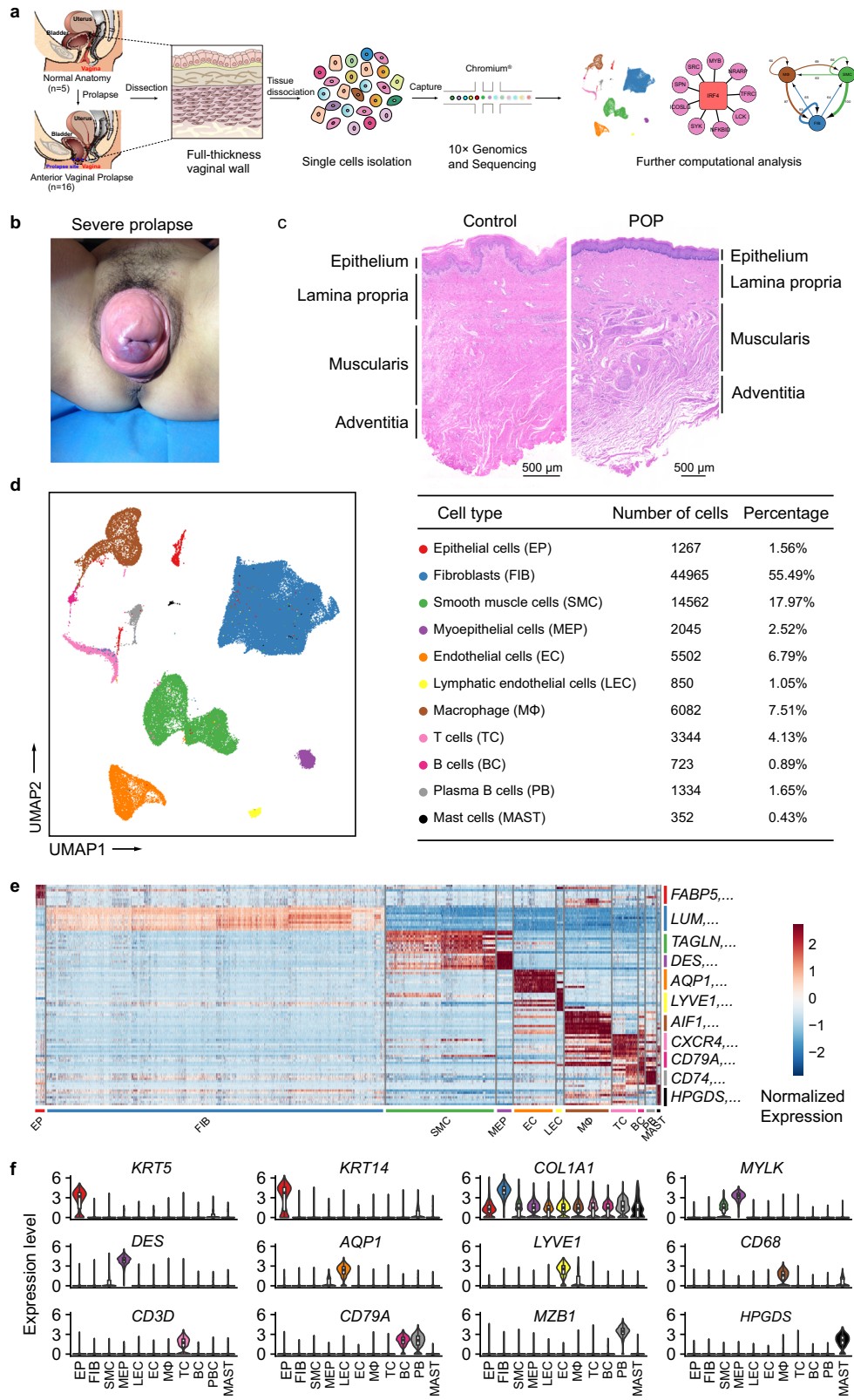

two most abundant cell types were fibroblasts (55.49%) and smooth muscle cells (17.97%), consistent with the known cellular composition of connective tissues. Notably, we identified 5 immune cell types, among which macrophages were the third most abundant cell type. Then, we identified highly expressed genes in each cell type and performed gene ontology (GO) on these genes. The results further confirmed the accuracy of the cell definitions (Supplementary Fig. 1d). To validate the presence of some representative cell types, we performed immunostaining for non-immune cells and flow cytometric analysis for immune cells (Supplementary Fig. 2b, c). Taken together, we revealed the cellular composition of the vaginal wall and provided a comprehensive representation of vaginal wall cells for further studies of prolapse.

**Fig. 1 Diverse cell types in the vaginal wall delineated by Single-cell RNA-seq analysis. a** Schematic of tissue dissociation, cell isolation, sequencing, and downstream bioinformatics analysis. **b** Representative pictures of patients with anterior vaginal prolapse. **c** Representative H&E staining of the vaginal wall in control and POP samples (control, three patients; POP, four patients). **d** UMAP plots of the major vaginal wall cell populations. Each point depicts a single cell, colored according to cell types (left). The chart showing the number and percentage of each cell type (right). **e** Heatmap showing the relative expression of top 10 genes in each cell type. The canonical markers for each cell type are color-coded and shown on the right. **f** Violin plots displaying the expression of canonical markers for each cell type (control, $n = 5$ patients; POP, $n = 16$ patients). The horizontal line within each box represents the median, and the top and bottom of each box indicate the 75th and 25th percentile. Two-sided Wilcoxon rank-sum test was applied to test the significance of the gene expression with $p$-value < 0.05. EP epithelial cell, FIB fibroblasts, SMC smooth muscle cells, MEP myoepithelial cells, EC endothelial cells, LEC lymphatic endothelial cells, MΦ macrophages, TC T cells, BC B cells, PB plasma B cells, MAST mast cells.

**Table 1 Statistical analysis of clinical characteristics of participants.**

| Variable | POP group ($n = 16$) | Control group ($n = 5$) | $p$-value |
|---|---|---|---|
| Age, mean ± SD, years | 65.25 ± 8.40 | 59.80 ± 4.44 | 0.185[a] |
| Body mass index, mean ± SD, kg/m$^2$ | 24.88 ± 2.62 | 25.49 ± 2.60 | 0.654[a] |
| Gravidity, median (interquartile), per child | 3 (2–4.75) | 2 (1.5–4) | 0.500[b] |
| Parity, median (interquartile), per child | 2 (1–2) | 1 (1–1) | 0.040[b] |
| Postmenopausal[d], $n$ (%) | 16 (100.0%) | 5 (100.0%) | – |
| Time since menopause, mean ± SD, years | 15.25 ± 9.38 | 9.20 ± 6.14 | 0.195[a] |
| Hormone replacement therapy, $n$ (%) | 1 (6.3%) | 0 (0.0%) | 1.000[c] |
| Chronic cervicitis or vaginitis history, $n$ (%) | 0 (0.0%) | 0 (0.0%) | – |
| History of malignancy, $n$ (%) | 2 (12.5%) | 2 (40.0%) | 0.228[c] |
| Smoking habit, $n$ (%) | 0 (0.0%) | 0 (0.0%) | – |
| Hypertension, $n$ (%) | 3 (18.8%) | 2 (40.0%) | 0.553[c] |
| Diabetes mellitus, $n$ (%) | 2 (12.5%) | 0 (0.0%) | 1.000[c] |
| Immune disorders history[e], $n$ (%) | 0 (0.0%) | 0 (0.0%) | – |

Descriptive data are given as numbers (%), means ± standard deviations, or medians (interquartile ranges).
[a]Two-tailed unpaired Student $t$-test of the *age, body mass index,* and *time since menopause.*
[b]Mann–Whitney $U$test (two-sided) was used to compare differences in different groups.
[c]A two-tailed χ2 test was used to compare differences in categorical variables. Boldface entries indicate $p \leq 0.05$ (statistically significant). Statistical analysis was performed using the software package SPSS (Version 25.0, SPSS Inc., Chicago, Illinois, USA).
[d]*Postmenopausal* was defined as at least 1 year after the cessation of menses.
[e]Immune disorders history included a history of asthma and autoimmune diseases such as systemic lupus erythematosus, rheumatic disease, or osteoarthritis, etc.

To delineate percentage changes in the cellular composition in POP, we compared scRNA-seq profiles between control and POP samples in accordance with a previous method (see "Methods")[32]. The cell types with altered proportions in the prolapsed vaginal wall are shown in Supplementary Fig. 2d. Globally, the proportion of epithelial cells was increased in the prolapsed vaginal wall, indicating that epithelial cells may show hyperplasia or hyperkeratinization, as previously reported[22]. Conversely, the proportion of smooth muscle cells in POP samples was decreased, partially consistent with histological observations indicating that prolapse induces atrophy of the muscularis. The proportions of fibroblasts and macrophages were similar between the POP and control samples, suggesting that abnormal gene expression in these cells may be more important than changes in their proportions in the pathological mechanism of prolapse. Combined with the results obtained by immunostaining and fluorescence-activated cell sorting (FACS) (Supplementary Fig. 2e), these data support the idea that epithelial cells and smooth muscle cells are altered in the prolapsed vagina. Thus, the cell type-specific mechanism in POP needs to be explored.

**Cell type-specific aberrant gene expression in POP samples**. To simultaneously define gene expression changes at the global and cellular levels, we also performed bulk RNA-seq of POP and control samples in parallel. In the bulk RNA-seq samples, 1190 upregulated genes and 1355 downregulated genes were detected in POP samples (Fig. 2a and Supplementary Data 4). To further investigate the biological function of these genes, functional enrichment analysis was performed on the upregulated and downregulated genes (Fig. 2b and Supplementary Data 5). The results indicated that cornification and epidermal cell differentiation were specifically activated in POP, whereas the genes downregulated in POP were mainly enriched in cell chemotaxis, leukocyte migration, and so on. Taken together, these results indicate that dysregulation of multiple molecular functions may be related to vaginal wall prolapse in POP.

To explore specific aberrancies in the expression of molecules in each cell type in the prolapsed vaginal wall, we evaluated changes in the expression of representative reported POP-related genes in the cell layer (Fig. 2c). *COL9A1*, shown to be upregulated in bulk RNA-seq, was upregulated mainly in fibroblasts, whereas its expression was unchanged in other cells. Conversely, *CXCL1* gene expression was downregulated in most cell types and upregulated in macrophages and mast cells. These differences reflect cellular heterogeneity in gene expression changes, further suggesting that investigating gene expression changes in each cell type in the prolapsed vaginal wall in POP is important.

Next, to identify gene dysregulation in POP at the level of cell type specificity, we detected differentially expressed genes (DEGs) in each cell type between POP and control samples. Several hundred to thousands of DEGs were detected in each cell type (Supplementary Fig. 3a and Supplementary Data 6). The number of upregulated genes was higher than the number of downregulated genes in all cell types except for epithelial cells, lymphatic endothelial cells, B cells, plasma B cells and mast cells (Supplementary Fig. 3b). Notably, each cell type contained some cell type-specific upregulated and downregulated genes (Supplementary Fig. 3a). Functional enrichment analysis indicated that the terms ECM organization and immune reaction were enriched in DEGs in both non-immune and immune cell types (Fig. 2d, e).

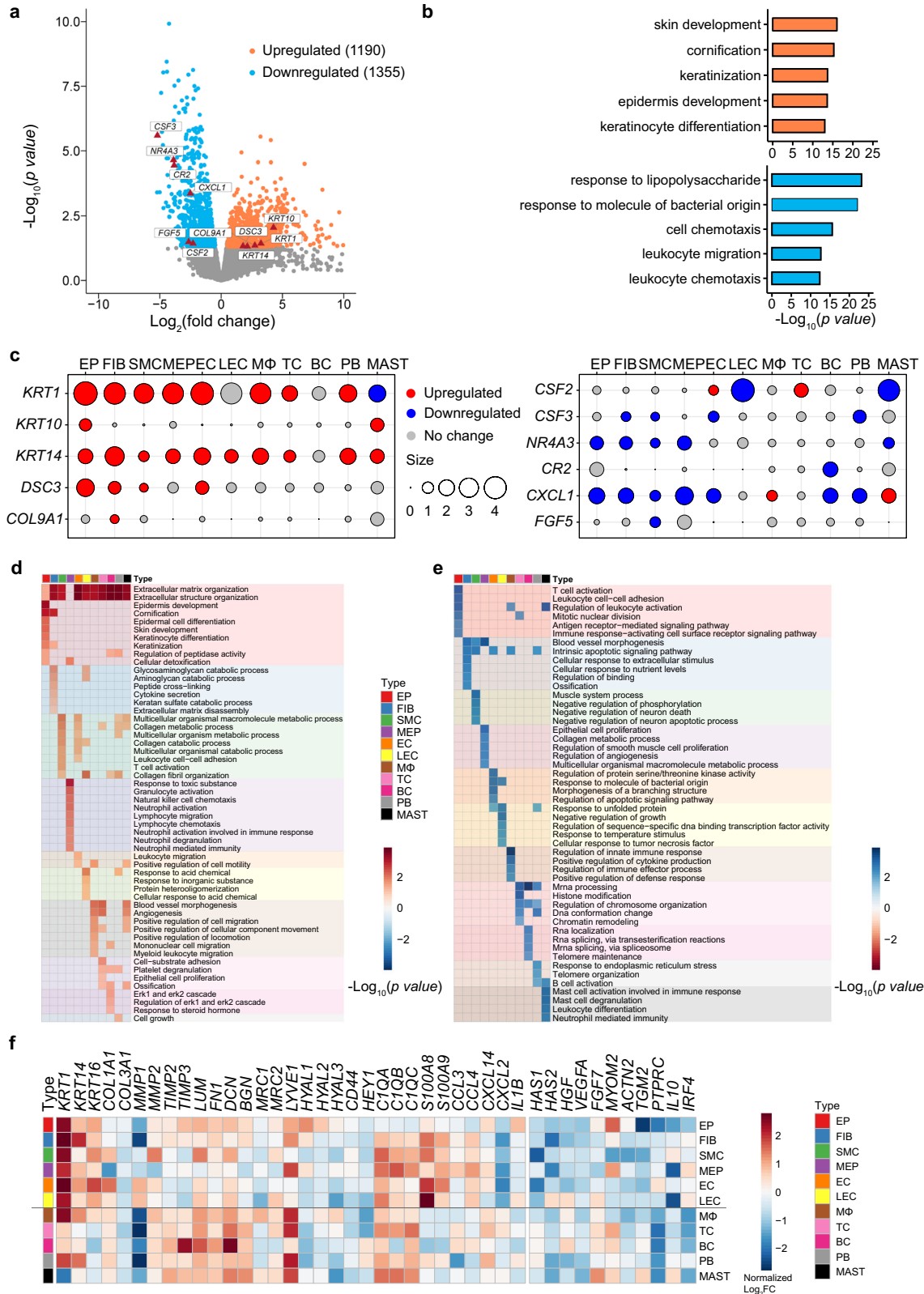

The enrichment of the term ECM organization and disassembly in most cell types suggested that diverse types of vaginal cells are widely involved in ECM dysregulation. In particular, the terms collagen catabolic process and collagen fibril organization were enriched in upregulated genes in smooth muscle cells from POP samples, while muscle system process was enriched in down-regulated genes, implying that the phenotypic function of smooth muscle cells might switch and play a vital role in collagen metabolism when prolapse occurs, consistent with previous studies reporting that smooth muscle cells undergo an aberrant switch from a contractile to a synthetic ECM phenotype and contribute to the distensibility and fragility of the prolapsed vaginal wall[33]. Another notable finding was the enrichment of immune reaction disorders in DEGs in multiple cell types from

**Fig. 2 Aberrant Gene Expression Profiles in cell type-specific manners in POP. a** Volcano plot showing the differentially expressed genes (DEG) in bulk RNA-seq data. **b** Bar plots displaying the gene ontology (GO) enrichment of up- or downregulated genes in POP (Control, $n = 5$ patients; POP, $n = 16$ patients). Fisher's exact tests (two-side) were performed. $p$-value < 0.05 was defined as statistically significant. **c** Dot plots showing the relative expression change of specific genes across different cell types. The size indicates the $\text{Log}_2\text{FC}$ values (POP/control). **d** and **e** Heatmap showing the representative gene ontology enriched in upregulated (**d**) or downregulated (**e**) genes in each cell type. The default fgsea algorithm on 1000 permutations with $p$-value < 0.05 was utilized. **f** Heatmap showing the relative expression for representative reported POP-related genes and other representative genes in each cell type in POP samples than that in control samples. EP epithelial cell, FIB fibroblasts, SMC smooth muscle cells, MEP myoepithelial cells, EC endothelial cells, LEC lymphatic endothelial cells, MΦ macrophages, TC T cells, BC B cells, PB plasma B cells, MAST mast cells.

POP samples. In immune cells from POP samples, the upregulated genes were enriched in immune response terms such as leukocyte migration and platelet degranulation. Moreover, the upregulated genes in fibroblasts and smooth muscle cells from POP samples were enriched in terms such as cytokine secretion and T cell activation (Fig. 2d, e). These data were consistent with those in previous reports indicating that immune cells participate in complicated interplay with non-immune cells and the ECM upon tissue injury[34,35]. However, these studies did not examine the prolapsed vaginal wall or suggest the vital role of the immune response in the prolapse process. Together, these results indicate that most cell types participate in ECM dysregulation and immune reaction disorder in the prolapsed vaginal wall during the prolapse process.

We then focused on the cell type-specific expression patterns of known genes related to POP, such as *COL1A1* and *COL3A1*, related to ECM components[36]; and *MMP1* and *MMP2*, related to matrix metalloproteinases[36–38] (Supplementary Fig. 4). Notably, beyond fibrillar collagens, the expression of ECM molecules such as glycoproteins and proteoglycans have extensive changes (Fig. 2f). For example, genes encoding ECM molecules (such as *FN1*, *LUM* and *DCN*) or receptors for cellular uptake of hyaluronan (HA) and collagen (such as *LYVE1* and *MRC2*) were extensively upregulated in most cell types in POP samples. Notably, two types of collagen endocytic receptors (*MRC1* and *MRC2*), HA degradation genes (e.g., *HYAL2*, *HYAL3*) and HA receptors (e.g., *LYVE1*), which regulate inflammation by transducing signals from the ECM, were upregulated in macrophages. Previous studies have reported that macrophages and fibroblasts are responsible for the participation of ECM or collagen modulators in various diseases[34,39]; therefore, we speculated that fibroblasts and macrophages may play vital roles in ECM dysregulation and immune disorder in POP.

**Abnormal transcription factor regulation in POP samples**. To assess the expression status of transcription factors (TFs) in the normal vaginal wall and identify potential transcription factors modulating the differential expression of genes in POP samples, single-cell regulatory network inference and clustering (SCENIC) was performed. Through this approach, we predicted the cell type-specific transcription factors in the vaginal wall in normal samples. These transcription factors were active in specific cell types and regulated cell type-specific functions (Supplementary Fig. 5a). For example, the transcription factor *WT1* was active in fibroblasts, and genes regulated by *WT1* were enriched in terms related to actin filament behavior (Supplementary Fig. 5a, b). In addition, *TBX2*, regulating smooth muscle cell chemotaxis and fibroblast activation, was active specifically in smooth muscle cells.

As fibroblasts, smooth muscle cells, and macrophages presented more aberrant DEGs than other cell types and the most obvious relationship with ECM dysregulation and immune disorder, we further detected the abnormally activated transcription factors in these three cell types in POP samples (Fig. 3a–c). *HOXD11* was highly expressed in all three cell types in POP

samples, and genes regulated by *HOXD11* were mainly enriched in reproductive structure development and upregulated in POP samples. These results indicated that *HOXD11* might play an important role in governing the involvement of these three cell types in ECM organization and the prolapse process. Previous studies have reported that members of the *HOX* family, especially *HOXA* genes controlling the reproductive system and collagen metabolism, is required for embryonic development[40,41]. Moreover, *IRF4*, *IRF8* and members of the *FOS/JUN* family were highly expressed in fibroblasts, smooth muscle cells and macrophages, respectively, in POP samples. *IRF4* and *IRF8* have immune-specific regulatory roles, while the *FOS/JUN* family can enhance the inflammatory responses of macrophages. Target genes of these transcription factors were upregulated and enriched in functions, including regulation of leukocyte cell–cell adhesion, regulation of immune effector process, and neutrophil-mediated differentiation. In summary, we identified dysregulation of several important candidate transcription factors regulating DEGs and functions in POP.

**Determination of aberrant cell–cell communication patterns in POP samples**. To define the intercellular communication networks within the vaginal wall, we first investigated the expression of ligand–receptor pairs in each cell type in control samples (Supplementary Fig. 6a). Fibroblasts and endothelial cells participate in the highest level of cell–cell communication. GO enrichment analysis indicated that ligands were involved in ECM organization and regulation of chemotaxis and that receptors were enriched in leukocyte migration, cell-matrix adhesion, and cell chemotaxis (Supplementary Fig. 6b, c). Specifically, we analyzed the interactions of fibroblasts, smooth muscle cells, and macrophages with other cell types (Supplementary Fig. 6d). The ligand–receptor pairs between fibroblasts/smooth muscle cells and non-immune cells were enriched mainly in ECM organization, cell-matrix adhesion and other terms.

We then detected altered cell-cell communication in POP samples compared to control samples. In POP samples, the interactions of fibroblasts with other cell types were decreased and those of smooth muscle cells were increased (Fig. 4a), indicating weaker and stronger interactions, respectively, for these two cell types in POP. Most of the gained interactions for fibroblasts and other cell types were enriched in fibrinolysis; in smooth muscle cells, the gained interactions were enriched in HA metabolic process (Fig. 4b), consistent with the findings for genes with upregulated expression (such as *FN1*, *LUM*, and *LYVE1*). Moreover, interactions of macrophages with fibroblasts and smooth muscle cells were increased, implying enhanced interplay among these three cell types (Fig. 4a). The gained interactions between macrophages and other cell types were enriched in ECM organization and regulation of wound healing (Fig. 4b), indicating the involvement of macrophages in ECM regulation and the immune response in the prolapsed vaginal wall.

In addition, to demonstrate the crosstalk among these three cell types, we evaluated the expression patterns of ligand–receptor pairs among these three cell types. Intercellular communication of

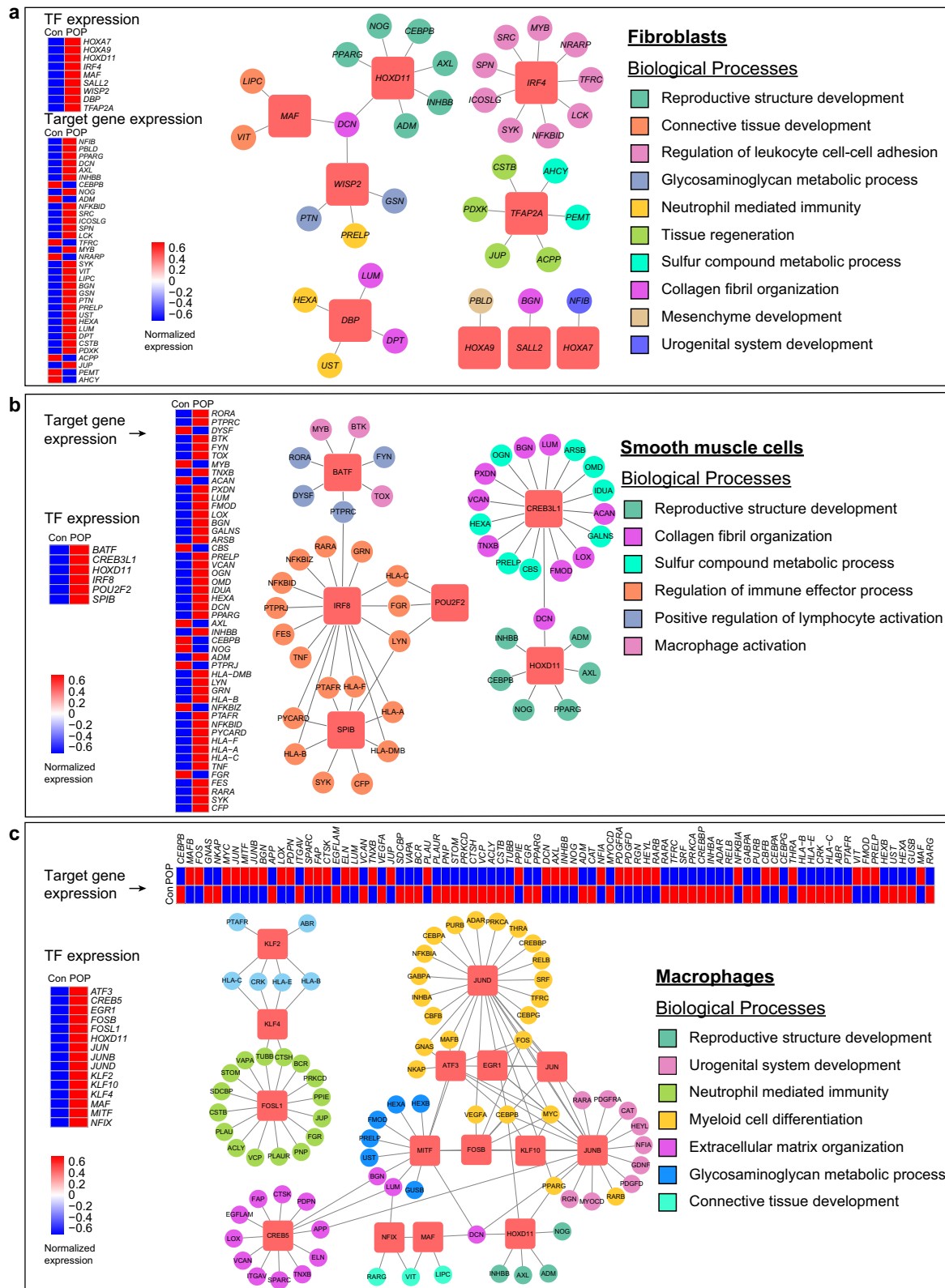

**Fig. 3 Single-cell network inference reveals candidate differential expression of transcription factors among major cell types. a–c** Representative upregulated TFs in the POP samples, which regulate ECM regulation and immune modulation and so on in fibroblasts (**a**), smooth muscle cells (**b**) and macrophages (**c**), respectively. The networks consist of several transcriptional factors and their target genes, color-coded by representative GO enrichment terms. The terms were listed on the right. The red square nodes represent TFs, and the round nodes represent target genes. Heatmaps of TFs and target genes expression were also shown on the left or the top. Con control, POP pelvic organ prolapse, TF transcription factors.

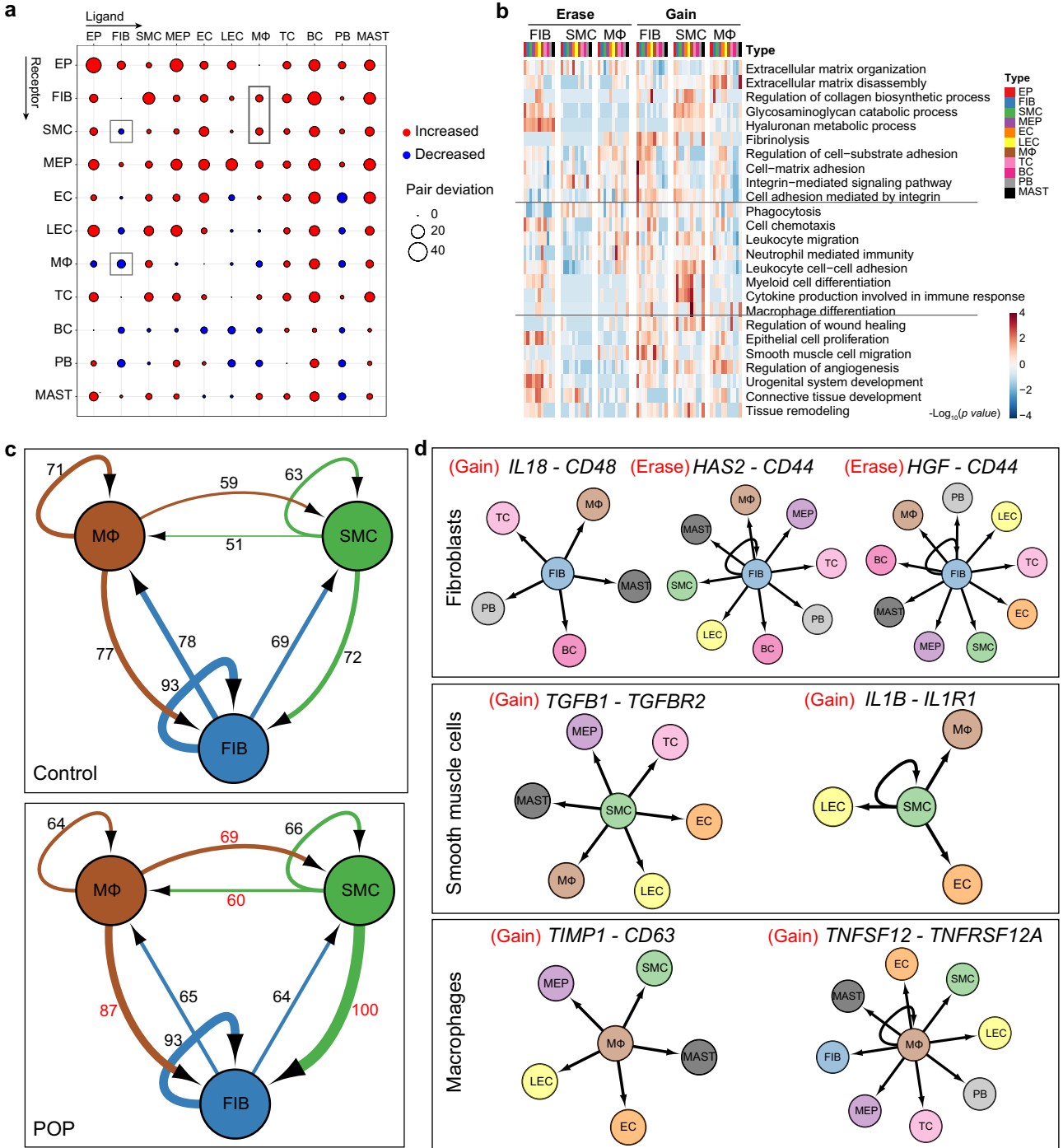

**Fig. 4 Global analysis of ligand–receptor interaction pairs. a** Dot plots depicts the changed numbers of putative ligand–receptor pairs in POP samples compared with control samples (red, increased; blue, decreased). **b** Heatmap displaying the gene ontology enrichment in the increased or decreased ligand–receptor pairs in fibroblasts, smooth muscle cells and macrophages. **c** Network visualization of ligand–receptor pair numbers among fibroblasts, smooth muscle cells and macrophages. The number of ligand–receptor pairs was shown. **d** Network visualization of specific pairs among different cell types between control and POP samples. EP epithelial cell, FIB fibroblasts, SMC smooth muscle cells, MEP myoepithelial cells, EC endothelial cells, LEC lymphatic endothelial cells, MΦ macrophages, TC T cells, BC B cells, PB plasma B cells, MAST mast cells.

smooth muscle cells and macrophages was increased in POP samples compared with control samples (Fig. 4c). Consistent with the results of DEG analysis, the gained interactions among these three cell types included cytokines and their receptors (Fig. 4d). For example, IL18-CD48 are pro-inflammatory cytokines, and their interaction was gained in fibroblasts and immune cells in POP samples. IL1B–IL1R1 interactions involved in inflammatory

activation were also gained in smooth muscle cells and some cell types. Notably, the differential interactions among these three cell types were involved in tissue regeneration and matrix organization. For instance, TGFB1–TGFBR2 interaction, which participates in tissue remodeling, was gained in smooth muscle cells and other cell types. TIMP1–CD63 interaction, which regulates ECM degradation and remodeling, was gained in macrophages and

non-immune cells. These data indicate that the interactions involved in immune regulation are widespread in fibroblasts, smooth muscle cells and macrophages and that the regulatory ability of smooth muscle cells and macrophages in tissue remodeling and matrix organization is enhanced upon prolapse. Herein, abnormal cell-cell communication patterns were shown to occur during vaginal wall prolapse, especially in fibroblasts, smooth muscle cells and macrophages, and the interactions are mainly related to ECM remodeling and immune modulation.

**Identification of cell subtypes and cellular alterations in POP samples**. We next explored the subtypes of some specific cell types. Fibroblasts are the most abundant cell type in the vaginal wall, and have historically been suggested to represent a heterogeneous population. However, the extent of fibroblast heterogeneity in POP is unexplored. In our samples, vaginal fibroblasts were detected, and 7 distinct subtypes were acquired by subclustering (Fig. 5a and Supplementary Fig. 7a). Most clusters were found in all samples (Fig. 5b, Supplementary Fig. 7b, Supplementary Data 7 and Supplementary Data 8). Proportionally, subtypes 1 and 3 were strongly enriched in POP samples, and subtypes 2, 6, and 7 were enriched in control samples (Supplementary Fig. 7c). We then evaluated the expression patterns of known genes related to POP, the collagen family, MMPs and TIMPs in the subtypes and found that the expression patterns of these genes differed in each subtype (Supplementary Fig. 7d). For example, *COL1A1, COL1A2, COL5A3, COL6A1, COL6A2,* and *COL6A3* were specifically highly expressed in subtype 7 but downregulated in subtype 5 in POP samples. As different collagens have different roles in the ECM, these differences suggest functional specialization of fibroblast clusters. Meanwhile, we also investigated the gene expression on transcription factors and ligand–receptor pairs of each subtypes compared with control samples. The important transcription factors and ligand–receptor pairs that were identified in cell clusters exhibited different expression patterns in each subtype (Fig. 5c). For example, *MRC1* and *MRC2*, DEGs related to collagen endocytosis, were highly expressed mainly in subtype 7, while transcription factors related to reproductive structure development (e.g., *HOXD11*) were expressed in subtype 1. Ligand–receptor interaction participants such as *HGF* and *CD44* were present in subtypes 7 and 5, respectively. Taken together, these results demonstrate that the contributions of fibroblast subtypes differ and further confirm the heterogeneity of fibroblasts.

Similarly, we classified smooth muscle cells into four transcriptionally distinct subtypes (Fig. 5d, e, Supplementary Fig. 8a–c, Supplementary Data 7 and Supplementary Data 8), which were present in all samples. Proportionally, subtype 3 was strongly enriched in POP samples, and subtype 2 was enriched in control samples (Supplementary Fig. 8d). Moreover, DEGs, transcription factors and ligand–receptor interaction pairs were also specifically expressed in some subtypes in POP samples compared to cell cluster (Fig. 5f).

Further, macrophages could be classified into five transcriptionally distinct subtypes that were present in all samples (Fig. 5g, h, Supplementary Fig. 9a, b, Supplementary Data 7 and Supplementary Data 8). Proportionally, subtypes 1 and 5 were strongly enriched in POP samples, and subtype 4 was enriched in control samples (Supplementary Fig. 9c). Most DEGs and transcription factors were highly expressed mainly in subtype 5, while the interactions of ligand–receptor pairs such as TNFSF12-TNFRSF12A, participants in another ligand–receptor pair, were present in subtypes 1, 3, and 5 and subtypes 3 and 4, respectively (Fig. 5i). Macrophages are conventionally polarized into M1 (inflammatory) and M2 (anti-inflammatory/phagocytic)

phenotypes[42], which perform different functions in normal and pathological processes—especially the M2 phenotype, which is consistently involved in tissue remodeling and wound healing[43]. Notably, M2 macrophages were proportionally increased in POP samples (Supplementary Fig. 9d). To identify the functions of the two different phenotypes, we performed GO enrichment analyses and revealed that phagocytic macrophages were mainly related to the response to transforming growth factor beta and collagen fibril organization (Supplementary Fig. 9e).

Finally, to provide fundamental data for future work and transcriptome profiles for analysis, we performed clustering analysis for epithelial cells, endothelial cells, and T cells in vaginal walls and detected five, three and three subtypes, respectively, of these three cell types (Supplementary Fig. 10a–f). Among endothelial cells, venule endothelial cells, capillary endothelial cells and arterial endothelial cells were identified (Supplementary Fig. 10c, d). Among T cells, central memory T cells, effector memory CD8$^+$ T cells and cytotoxic CD8$^+$ T cells were found (Supplementary Fig. 10e, f).

## Discussion

The prevalence of symptomatic POP, which interferes with women's physical and mental well-being, is 9.6% in women aged >20 years and 15.7% in women aged >50 years in China and increases with advancing age[3,4]. However, its pathophysiology is not completely clear. Thus, it is highly desirable to explore the molecular mechanisms pivotal for vaginal wall prolapse in POP. In this study, we presented the first single-cell survey of various cell types in the anterior vaginal wall of POP patients and elucidated the cell type composition and cell type-specific gene expression signatures in the prolapsed vaginal wall, providing insights into the mechanisms related to vaginal prolapse in POP. Notably, 11 cell types, including 6 conventional cells in connective tissue and 5 types of immune cells, were identified. Importantly, we defined the transcriptional signatures and the DEGs in each cell type and discovered that ECM organization and immune and inflammation reactions were upregulated in most cell types, suggesting synergistic effects of vaginal cell types in POP. Furthermore, we identified alterations in transcription factors and cell-cell communication in POP. Taken together, these observations provide insights into POP and identify targets explaining the pathological processes of POP and developing related therapeutic strategies.

Here, we obtained anterior vaginal wall samples from both patients with POP and control individuals and successfully mapped the first single-cell transcriptome atlas, providing high-quality data to reveal in-depth POP-related alterations in gene expression in each cell type and subtype at the single-cell level. The cell types in POP samples were the same as those in control samples, while the major alterations were seen in gene expression in each cell type. Although important biological processes and putative candidate genes most likely linked with POP development, such as ECM remodeling and related genes, have been revealed via bulk RNA-seq[10,12,20], due to cell type heterogeneity, bulk RNA-seq results might not identify whether these changes are intrinsic molecular changes or simply reflect changes in the proportions of cell types. Our study confirmed the gene expression changes in each cell type and found that some genes were dysregulated in most cell types; moreover, some cell type-specific upregulated and downregulated genes were identified. Notably, ECM or structural organization was upregulated in most cell types. In addition, immune reaction disorder and immune cell dysfunction were identified in the prolapsed vaginal wall of POP. Immune and inflammatory cells can engage in complex interplay with resident non-immune cells and the ECM of tissue during

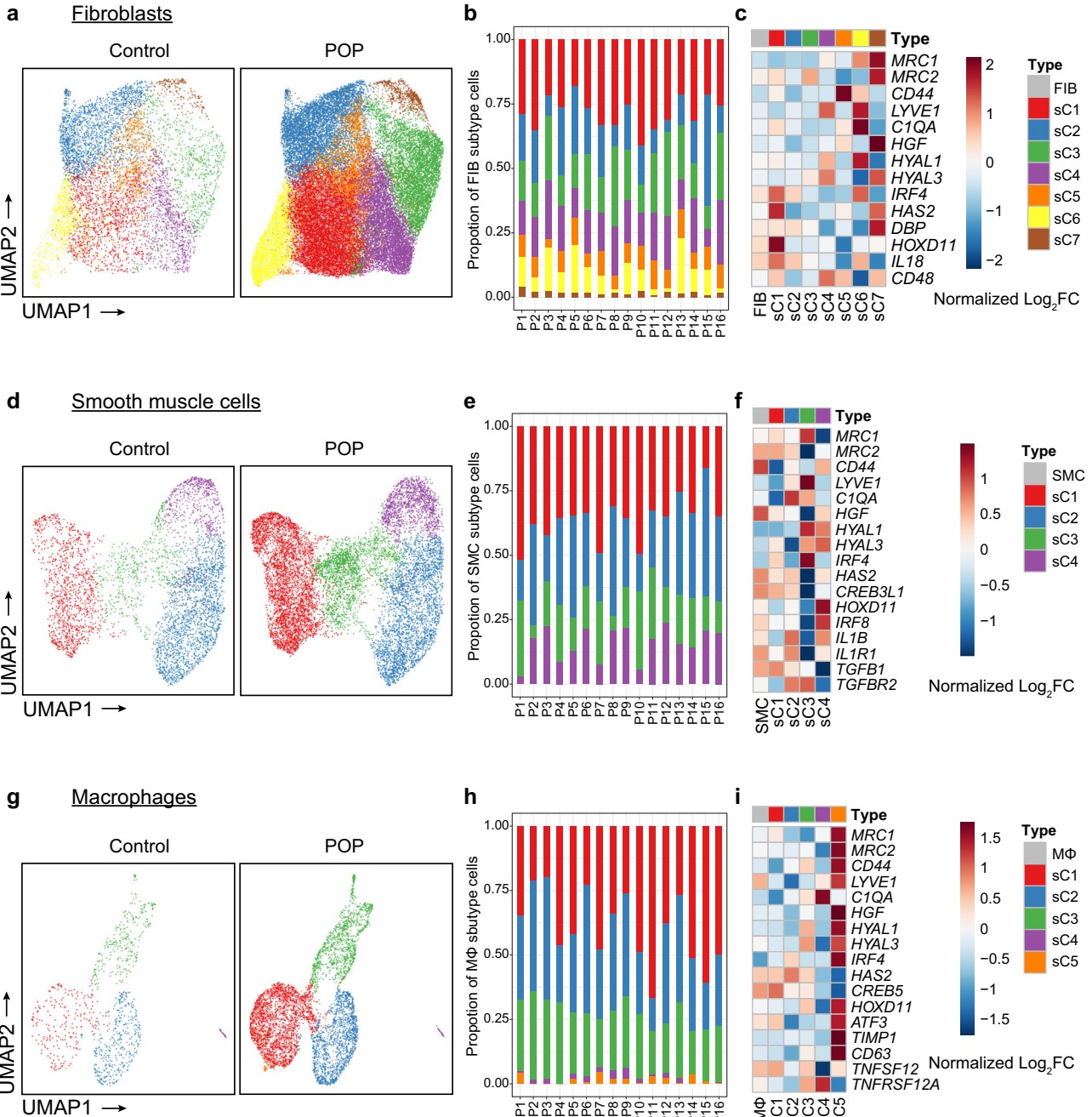

**Fig. 5 Subclustering of major cell types in the vaginal wall reveals cellular heterogeneity. a** UMAP plot showing the distribution of seven distinct fibroblasts subtypes in control and POP samples. **b** Bar plots showing the percentage of seven fibroblast subtypes in each POP patient ($n = 16$ patients). **c** Relative expression of representative DEGs, TFs and ligand–receptor pairs among seven subtypes in POP samples than that in control samples. **d** Subclustering of smooth muscle cells further identified four distinct subtypes. **e** Bar plots showing the percentage of four smooth muscle cells subtypes in each POP patient ($n = 16$ patients). **f** Relative expression of representative DEGs, TFs and ligand–receptor pairs among four subtypes in POP samples than that in control samples. **g** Subclustering of macrophages further identified five distinct subtypes. **h** Bar plots showing the percentage of five macrophages subtypes in each POP patient ($n = 16$ patients). **i** Relative expression of representative DEGs, TFs and ligand–receptor pairs among five subtypes in POP samples than that in control samples. P, POP.

tissue injury[35,44]. The ECM and the immune system are intertwined, which may promote ECM repair and regeneration or exacerbate these conditions through increased and sustained inflammation. However, this process has not been reported in the prolapsed vagina. Our work revealed the insight into ECM dysregulation in the prolapsed vagina.

Considering that transcription factors regulate gene expression, we identified multiple previously unreported transcription factors

related to POP, such as *HOXD11*, *CREB3L1*, *IRF4*, and *IRF8*. The transcription factor *HOXD11*, which targets genes involved in reproductive structure development, was upregulated in fibroblasts, smooth muscle cells and macrophages in POP samples compared with control samples. In addition, we identified transcription factors related to collagen fibril organization (such as *CREB3L1*) that were upregulated in smooth muscle cells in POP samples. Strikingly, immune response-related transcription

factors were widely upregulated in both fibroblasts and smooth muscle cells, implying the potential synergistic effect of fibroblasts and smooth muscle cells on immune reactions when prolapse.

Cell–cell communication is important for various biological processes. We found that intercellular communication was altered in POP. Interactions between smooth muscle cells, fibroblasts and macrophages in POP samples were increased compared with those in control samples. In addition, interactions regarding immune regulation and cytokine secretion were gained in fibroblasts and smooth muscle cells. Moreover, tissue remodeling-related interactions such as the TGFB1–TGFBR2 interaction were gained between smooth muscle cells and some cell types, indicating that these cells participate in ECM and tissue modeling. The phenotypic switch from smooth muscle cells to myofibroblasts could be the underlying cause of structural changes in the muscularis[33].

Overall, this study provides comprehensive single-cell transcriptome atlas for deciphering the gene expression landscapes of heterogeneous cell types in the anterior vaginal wall of POP and broadens our understanding of cell identities and cell type-specific gene alterations in POP. The single-cell transcriptome atlas from POP and control samples presented here can be a great resource for POP research. Moreover, we revealed the critical genes and key transcription factors that might coordinately regulate POP development. In addition, we discovered that altered cell-cell communication may result in the disorder of normal cellular function. Thus, these findings are potentially valuable for understanding the critical molecular mechanism underlying POP and improving current preventative and therapeutic strategies for this disorder.

## Methods

**Patient samples**. All tissue samples used for this study were obtained with informed consent from all patients and the procedures in this study were reviewed and approved by the Ethics Committee of Peking Union Medical College Hospital (JS-1605). Postmenopausal women undergoing hysterectomy surgery for POP and other benign indications were enrolled. Participants with Pelvic Organ Prolapse Quantification (POP-Q) stage III or IV in the anterior compartment with or without other compartments prolapse were included in the study under the diagnosis of the same experienced expert. Exclusion criteria were prior pelvic reconstruction surgery, chronic pelvic inflammation, chronic debilitating disease, autoimmune and connective tissue disorders or cancer. Women undergoing hysterectomy for benign gynecological reasons and without prolapse were enrolled as control groups. All patients were non-smokers with similar parity. Age, parity and other characteristics of patients were listed in Supplementary Data 1. Full thickness (1 cm$^2$) vaginal wall tissue biopsies were harvested from the pericervical region of the anterior vaginal cuff after hysterectomy in the controls, and from the prolapsed vaginal wall in the POP samples.

**Tissue dissociation and preparation of single-cell suspensions**. Fresh isolated tissues were immediately placed in ice-cold DMEM (Wisent, 319-005-CL) with 1% fetal bovine serum (FBS, Wisent, 086-150) and then transported on ice to preserve viability. Vaginal wall tissues were washed 2-3 times with PBS and dissected on ice to smaller pieces, then transferred to 10 mL digestion medium containing 1 mg/mL Collagenase Type I (Gibco, 17100-017), 2 mg/mL Dispase II (Sigma-Aldrich, D4693), 0.5 mg/mL Elastase (Solarbio, E8210) and 1 unit/mL DNase I (NEB, M0303S) in PBS with 1% FBS. The tissue was enzymatically digested at 37 °C with a shaking speed of 70 r.p.m for about 60 min. The dissociated cells were collected at interval of 20 min to increase the cell yield and viability. Cell suspensions were filtered using a 40-μm nylon cell strainer (Corning, 352340) and dissolved by the RBC lysis buffer (Invitrogen, 00-4333-57) with 1 unit/mL DNase I to remove red blood cells. Dissociated cells were washed with PBS containing 0.04% Bovine Serum Albumin (BSA; Sigma-Aldrich, B2064) and centrifuged at $500 \times g$ for 5 min to obtain the cell pellet. The cell viability was determined by Trypan blue (Invitrogen, T10282) staining and then cells were suspended in PBS with 0.04% BSA at a density of about $1 \times 10^6$ cells/mL and kept on ice for single-cell sequencing.

**10× Single-cell library construction and sequencing**. Single-cell suspensions were converted to barcoded scRNA-seq libraries according to standard protocols of the Chromium single-cell 3' kit in order to capture 5000 to 10,000 cells/chip position (V2 chemistry). All the remaining procedures including the library construction were performed according to the standard manufacturer's protocol. The libraries were applied to pair-end sequencing with read lengths of 150nt on Illumina HiSeq Xten platform.

**scRNA-seq data processing and determination of the major cell types**. Droplet-based sequencing data were mapped to the GRCh37 human reference genome through Cell Ranger Single-Cell Software Suite (version 2.1.0, 10x Genomics) to generate digital gene expression matrices. The data from all samples were read into the Seurat R package (version 3.1.2.9010)[45] for the further processing. Firstly, data filtering was conducted by retaining cells expressed 500 and 3500 genes inclusive, and had mitochondrial content less than 10 percent. Each library was scaled by library size and log-transformed (using a size factor of 10,000 molecules per cell). The top 2000 highly variable Genes (HVGs) were identified through the function "FindVariableGenes." In order to exclude multiple captures, which is a major concern in microdroplet-based experiments, DoubletFinder (version 2.0.2)[46] was employed to remove top $N$ cells with the highest pANN score for each library separately, where $N$ represents the doublet rates in HiSeqXten platform. Then all the datasets were integrated using the "FindIntegrationAnchors" and "IntegrateData" function in Seurat. Merged data were scaled to unit variance and zero mean. The dimensionality of data was reduced by principal component analysis (PCA)[47]. A K-nearest-neighbor graph was constructed based on the euclidean distance in PCA space using the "FindNeighbors" function and Louvain algorithm was applied to iteratively group cells together by "FindClusters" function with optimal resolution on the optimal principal components. Visualization was achieved by the UMAP[48]. Finally, specific markers in each cluster were identified by the "FindAllMarkers" function and clusters were assigned to known cell types using the canonic markers. In addition, we filtered two clusters for their tiny cell number. Subclustering for major cell types was performed in the same way.

**Differential gene expression analysis**. Since droplet-based sequencing technique only could capture a portion of the transcripts in any cell, which causes many transcripts to be undetected and induces an excess of zero read counts, furthermore, it can also make the difference of gene expression even within cells of the same type, all these problems leading to power issues in differential expression (DE) analysis for single-cell RNA sequencing (scRNA-seq), especially the lowly expressed genes. To address this issue, we implied the method that summed the raw UMI counts from matrices for each gene in each cluster over groups of twenty cells and treated these "pseudobulks" as technical replicates for further differential gene expression analysis[49]. Differentially expressed genes between different groups were determined by R package edgeR (version 3.18.1)[50] with |Log$_2$-fold change| > 0.5 and $p$-value < 0.05 as thresholds. Gene ontology enrichment analysis was performed by clusterProfiler R package (version 3.13.0)[51]. Gene set enrichment analysis was carried out by fgsea R package (v1.8.0) with biological procession gene sets (c5.bp.v6.2.symbols.gmt); The default fgsea algorithm on 1000 permutations with $p$-value < 0.05 was utilized[52].

**SCENIC analysis**. The SCENIC analysis was performed as described in previous study[53]. Briefly, the expression matrix of raw UMI counts from each group (each cluster or each sample) were used as input into the command-line interface functions of pySCENIC (version 0.9.15)[53]. After default data filtering, GRNboost2 (arboreto 0.1.5) method[54] was utilized to generate gene regulatory networks. The cisTarget Human motif database v9 of regulatory features 10 kb centered on the TSS were used to identify the enriched motifs via "ctx" function, and individual cells were scored for motifs using the "aucell" function.

**Cell–cell communication analysis**. In order to explore cell-cell communication networks via ligand–receptor interactions, we employed the similar analysis proposed by Vento-Tormo et al.[55]. In our analysis, we initially filter out all ligand–receptor pairs expressed in less than 10% of cells in each cell type population based on a public repository of ligands, receptors and interactions database CellPhoneDB. To identify the significant cell–cell interaction, we performed permutation tests between two cell types mediated by a specific ligand–receptor pair based on the mean gene expression of ligand from one cell type and the corresponding receptor from another cell type, and $p$-value < 0.01 was considered statistically significant. This procedure was performed between all pairs of cell types. Finally, the number of the significant ligand–receptor pairs represent the weights of the edges between each pair of cell types.

**Differential proportion analysis**. Differential proportion analysis was performed based on the ratio change across different conditions. First, we got the proportion of each cell type or subtype by dividing the numbers of cells by the total number of cells in different groups. Then, the Log$_2$-fold change was calculated between control and POP samples and |Log$_2$-fold change| > 0.5 was considered as threshold for significant change.

**Bulk RNA-seq library preparation and sequencing**. After quality control and quantification of the RNA obtained from control and POP vaginal wall tissues,

sequencing libraries were generated using KAPA Stranded mRNA-Seq Kit for Illumina® Platforms according to the manufacturer's recommendations. The indexed libraries were sequenced on an Illumina Hiseq 2500 platform and 100 bp/150 bp paired-end reads were generated.

**Bulk-seq analysis.** The quality of raw paired-end sequencing reads was checked by FastQC (version 0.11.5). Genomic alignment was performed using HISAT2 (version 2.0.5)[56] aligner to human reference genome (GRCm37/hg19; Ensembl version 72). FeatureCounts (version 1.6.0)[57] was employed to calculate the read counts of per gene. Differentially expressed genes between samples were identified by edgeR R package (version 3.18.1)[50] with |Log$_2$-fold change| >0.5 and $p$-value < 0.05 as thresholds.

**Histology analysis.** Vaginal wall tissues were harvested for histology analyses according to the following standard procedures. Briefly, tissues were fixed in formalin and dehydrated, then embedded in paraffin and cut into sections with a thickness of 5 μm. Hematoxylin and eosin (H&E) stain were performed with standard protocols to examine the tissue morphology change. Immunohistochemistry and immunofluorescence staining were performed to characterize main cell types. Paraffin-embedded tissue sections were deparaffinized and rehydrated in graduated alcohol, then treated in 0.1 M sodium citrate buffer and heated for 30 min for antigen retrieval. After cooling down, the endogenous peroxidase activity was blocked by 3% (vol/vol) $H_2O_2$, and then the slides were incubated with primary antibodies respectively. Parallel controls were run with PBS. After incubation overnight, the sections were washed with PBS and then subjected to the secondary antibodies. For immunohistochemistry staining, the sections were incubated by HRP-linked secondary antibodies and visualized with diaminobenzidine (DAB) staining. Counterstaining was performed with hematoxylin. For immunofluorescence staining, the sections were incubated by DyLight 488/549 AffiniPure-conjugated secondary antibodies (EarthOx) and counterstained with DAPI (Solarbio, s2110). The following primary antibodies were used for immunostaining: alpha smooth muscle actin (α-SMA, Abcam, ab32575, 1:300 for IHC; 1:200 for IF), smooth muscle protein 22-alpha (SM22-alpha, SM22, or Transgelin, Abcam, ab10135, 1:500), cytokeratin 14 (KRT14, Abcam, ab7800, 1:100), von willebrand factor (VWF, Abcam, ab201336, 1:500), vimentin (VIM, Abcam, ab92547, 1:300). The percentages of positive regions were quantified using Image J. Control, $n = 3$ patients; POP, $n = 4$ patients.

**Flow cytometry.** Fresh vaginal wall tissues were dissociated using the single-cell preparation procedures. After enzymatically digestion and washing, single cells were resuspended in 100 μL Stain Buffer (Biolegend, 420201) freshly prepared with 3 μL of each antibody. Cells were stained for 30 min on ice, then washed with PBS and resuspended at $1 \times 10^6$ cells/mL. T cells, B cells, and macrophages were investigated using the following antibody panels: CD3 (Biolegend, 300316, 3:100), CD19 (Biolegend, 302215, 3:100; 302218, 3:100), CD68 (Biolegend, 333806, 3:100). Flow sorting was performed on a BD FACS Aria II flow cytometer, and data were analyzed using FlowJo software (v.10.0.7, BD Biosciences). Control, $n = 4$ patients; POP, $n = 4$ patients.

**Statistical analysis.** Data are presented as the means ± SEM. $p$-values were calculated using a two-tailed Student's $t$-test in histological analysis and FACS analysis. The statistical analysis of clinical characterization was performed using SPSS 25.0 and mentioned in the legend. $p$-value < 0.05 was considered statistically significant. Two-side Wilcoxon rank-sum test with $p$-value < 0.05 was performed in Supplementary Data 2 and Supplementary Data 3. Likelihood-ratio test with $p$-value < 0.05 was performed in Supplementary Data 4 and Supplementary Data 6. Fisher's exact tests with $p$-value < 0.05 was applied in Supplementary Data 5.

**Reporting summary.** Further information on research design is available in the Nature Research Reporting Summary linked to this article.

## Data availability

Raw sequencing data and processed data are available through the NCBI Gene Expression Omnibus (GEO) under accession number "GSE151202" and at the Genome Sequence Archive (GSA) with accession number "HRA000136". These data have been deposited in GSA under project PRJCA002344. All other data supporting the findings of this study are available within the article and its Supplementary Information files or from the corresponding authors upon reasonable request. Source data are provided with this paper.

## Code availability

All analysis codes are available on Github at https://github.com/zqyhyunbin/POP[58].

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

## Acknowledgements

This project was funded by CAMS Initiative for Innovative Medicine (2017-I2M-1-002), the National Key R&D Program of China, Stem Cell and Translational Research (2019YFA0110901), Key Research Program of Frontier Sciences, CAS (QYZDY-SSW-SMC027, ZDBS-LY-SM013), the K. C. Wong Education Foundation, Shanghai Municipal Science and Technology Major Project (2017SHZDZX01).

## Author contributions

Y.G.Y. and L.Z. conceived this project. Y.Q.L. performed single-cell isolation. Y.Q.L., Y.D.M., Y.Z., and C.C.M. performed immunostaining analyses. Y.Q.L. and W.M. performed FACS analyses. Q.Y.Z. and B.F.S. performed bioinformatics analyses. H.H.S., Z.J.S., J.C., and L.Z. recruited patients. Y.Q.L., Q.Y.Z., B.F.S., Y.G.Y., and L.Z. wrote the manuscript.

## Competing interests

The authors declare no competing interests.

## Additional information

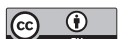

