## [Peer Review File · Nature Communications]

Reviewers' Comments:

Reviewer #1:

Remarks to the Author:

Li et al. in their manuscript entitled "Single-cell transcriptome profiling of the vaginal wall in women with severe anterior vaginal prolapse" used single-cell RNA-seq to construct a transcriptome atlas of 81,026 individual AVP cells and control samples and identified 11 cell types in 16 women with stage III-IV anterior vaginal prolapse compared to 5 controls. This is the first study to examine single cell gene expression changes. They found dysregulation of the extracellular matrix and immune-complex reactions, including associated transcription factors and cell-cell communication. This paper represents a resource for enhanced understanding of cellular heterogeneity in women with pelvic organ prolapse. I have listed a number of comments on the supplementary tables as this is the data of the study and allows others to reproduce the results generated if they so wish.

- 1) Abstract – lines 30-31 are essentially repeated in lines 32-33.
- 2) Line 162 states that fibroblasts altered in POP tissue, although the proportion is the same as in the control tissue. Based on the previous sentences it is unclear why fibroblasts are altered in the prolapse samples. Some comments on the immunostaining and fluorescence-activated cell sorting differences might account for the alterations in the fibroblasts.
- 3) The supplementary tables are very important to make sure that they are double checked for errors. There are several tables (e.g., Supplementary Tables 4 and 5) where gene names are cut-off because they extend beyond the allotted space. It would be most helpful to use a spreadsheet program such as Excel to display the tables so that they can be sorted and full names displayed rather than using a PDF format.
- 4) In the clinical characteristics supplementary table (Table 1), 21 POP cases are listed and 12 controls are listed, but only 16 cases are described in the text and 5 controls
- 5) While BMI is a risk factor for POP, the major risk factors for POP are not listed for the individuals who are part of your study including vaginal birth, family history of POP, and birth weight of infants. Please include this additional information if it is available.
- 6) The supplementary bulk DEG table (Table 4) contains Gene IDs rather than Gene names. It would be much more useful if it contained Gene names in addition to Gene IDs.
- 7) The supplementary Table 5 includes gene enrichment descriptors that were not included in the manuscript (Figure 2) such as skin development that had the highest p-value. While skin development is not typically thought to be associated with POP, all terms should be included to be unbiased to the generated results. Also, note that some of the values in this supplementary table are cut-off and cannot be read in full. Again, a spreadsheet format would solve this problem.
- 8) Supplementary Table 6 contains the DEGs for the 11 clusters. The clusters are not provided and note that one of the genes listed is 3-Mar, likely an inadvertent spreadsheet change.

Reviewer #2:

Remarks to the Author:

– Summary

In this paper Li et al use single cell transcriptomics to investigate severe anterior vaginal prolapse. The authors contrast data between AVP cases and tissue from matched healthy controls. This allows them to deconvolve known molecular and histological phenotypes into cell type specific changes in distribution and molecular activity. All analysis is sound and on the refreshingly simpler side compared to many other paper making use of scRNA-seq technology. The paper is straightforward and clearly contributes to knowledge about AVP. It sets up a next step of making use of the identified cellular and molecular changes to study causes of AVP. Only some minor issues about reporting need to be addressed.

— Minor issues

In figures it is not clear which abbreviation corresponds to which cell type. A legend would be helpful, either in the figure, in the text, or in the figure legend.

When using scRNA-seq to study differences in proportions of cell types the authors should report how large the differences are, and a certainty measurement such as a P-value. In the first instance differential cell type proportions are studied, proportions are quantified and validated using IHC and FACS. But such measurements are not done in the analysis of cell subtypes. Quantifications of differences of proportions are needed for Fig.S7, Fig.S8, and Fig.S9. It would also be good to add this to Fig.S2 even though quantification is performed using other measurements.

Reviewer #3:

Remarks to the Author:

This is a well-written paper on scRNA-seq analysis of anterior vaginal wall tissues from patients with prolapse compared to controls. My comments are listed below.

1. Much of the findings from this analysis is consistent with what is already published, for example, alterations in ECM, smooth muscle cell content, and immune/inflammatory pathways. The process of tissue keratinization in prolapse is also well documented since the tissues undergo keratinization from exposure to the outside environment and downward pressure. While this increases confidence that the scRNA-seq is yielding reliable data, it not novel.

2. What is novel is the finding of different sub-groups of fibroblasts and smooth muscle cells. However, this finding is not well developed in the manuscript. Figure 5 is confusing in that it is difficult to see how POP and controls differ in the distribution of these subgroups. Further characterization and verification of these subgroups would be of interest to the scientific community. scRNA-seq is a good methodology to investigate the rare sub-groups. These data may reveal interactive mechanisms that have not been considered.

3. Please provide a table with patient characteristics and statistical comparisons of these. differences in age, BMI, and parity can confound the data.

Revision Summary

1. Summary of major comments from the reviewers.

We want to thank reviewers for their critical and insightful comments. We have now performed further analyses and additional experiments to answer these concerns. These suggestions and our new experimental data further confirmed our original findings and improved the manuscript. The details of these revision results are summarized and listed in the following **Table 1**. Point-by-point responses to the editors and the reviewers' comments and suggestions are also included. The questions are italicized followed by our answers (Response) in which the red parts are the corresponding changes made in the manuscript. The corresponding changes in the manuscript and supplementary data are also highlighted by red color.

Table 1. Revision results for major comments from the reviewers.

Questions	Reviewers	Clarification on the original submission data	Performed experiments	New data supporting
1 Gene names in the supplementary tables are cut-off. It would be most helpful to use a spreadsheet program such as Excel to display the tables so that they can be sorted and full names displayed rather than using a PDF format. One of the genes listed is 3-Mar in Supplementary Table 6, likely an inadvertent spreadsheet change.	1#: Q3, 7, 8	We chose the inappropriate format in the submission system when uploading supplemental files. Now, we have re-uploaded the Excel tables.	N/A	We have revised them in Supplementary Tables 3, 4, 6.
2 Please include vaginal birth, family history of POP, and birth weight of infants in Patient's information.	1#: Q5	N/A	We have added vaginal birth, family history of POP and birth weight of infants in supplementary table 1.	We have added this additional information in Supplementary Table 1.

3	The supplementary bulk DEG table (Table 4) should have Gene names.	1#: Q6	N/A	N/A	We have newly added Gene names in Supplementary Table 4.
4	All GO terms should be included to be unbiased to the generated results in Figure 2.	1#: Q7	N/A	N/A	We have re-plotted the original results of GO enrichment (Revised Figure 2b).
5	In figures, it is not clear which abbreviation corresponds to which cell type.	2#: Q1	N/A	N/A	We have newly added annotations for abbreviation in all figure legends.
6	When using scRNA-seq to study differences in proportions of cell types the authors should report how large the differences are, and a certainty measurement such as a P-value.	2#: Q2	we employed the method from Ma's work ¹ as the reference. By calculating the Log ₂ FC between two groups ($ \text{Log}_2\text{FC} > 0.5$) to identify the cell types alteration along different condition, the value of $ \text{Log}_2\text{FC} $ represents the difference.	N/A	we have newly added the additional file with raw value of cell proportion for reference (Supplementary Table 7; Revised Figures s2, s7, s8 and s9.
7	Quantifications of differences of proportions are needed for Fig.S7, Fig.S8, and Fig.S9. Further characterization and verification of these subgroups would be of interest to the scientific community.	2#: Q2 3#: Q2	N/A	We have performed IF staining of specific markers for subtypes according to the gene markers in the	We have successfully validated four subtypes of fibroblasts, two subtypes of smooth muscle cells and two subtypes of macrophages for

				scRNA-seq results.	Fig.S7, Fig.S8, and Fig.S9 (Rebuttal Figures 1-4).
8	What is novel is the finding of different sub-groups of fibroblasts and smooth muscle cells. However, this finding is not well developed in the manuscript. Figure 5 is confusing in that it is difficult to see how POP and controls differ in the distribution of these subgroups.	3#: Q2	N/A	To better demonstrate the difference in the sub-groups, we visualized the subtypes via UMAP plot and showed the proportion change of each subtype between control and POP samples. We have also added the expression patterns of important DEGs, TF, ligand-recept or pairs in each cluster and subcluster.	We have added the UMAP projection of subtypes in POP and control samples. In addition, we also investigated the gene expression on transcription factors and ligand-receptor pairs of each subtypes compared with control samples. The important transcription factors and ligand-receptor pairs that were identified in cell clusters exhibited different expression patterns in each subtype (Revised Figure 5)
9	Please provide a table with patient characteristics and statistical comparisons of these differences in age, BMI, and parity can confound the data.	3#: Q2	N/A	We have performed analysis to investigate the differences of cell proportion and DEGs in	The results indicated that there was no significant difference of cell subtypes distribution and cell proportion in the two different

				subtypes between control and POP samples in different age, BMI and parity respectively. POP patients were divided into several groups according to the following rules: Age groups (<65 age, ≥65 age); BMI groups (18.5-24.9, ≥25); Parity groups (=1, >1) and then each group was compared with control sample, respectively.	groups (Age: <65 age vs ≥65 age, BMI: 18.5-24.9 vs ≥25; Parity: =1 vs >1). Compared with control samples, GO enrichments showed that DEGs in two different groups (Age: <65 age vs ≥65 age, BMI: 18.5-24.9 vs ≥25; Parity: =1 vs >1) exhibited similar trends that upregulated genes were involved in the extracellular matrix organization in most cases. While specifically, DEGs involved in one GO term such as extracellular matrix organization showed slightly different expression level. For example, although THBS4, LYVE1 and MMP11 were upregulated in both two age groups and HAS2 was downregulated in both two age groups, the relative
--	--	--	--	---	---

					expression level was different. The same expression trends but the different expression level in a certain gene are partly in consistent with the influence of age, BMI and parity which are associated risk factors of POP (Rebuttal Figures 4-7).
--	--	--	--	--	---

2. Point-by-point responses to the editor

Please check the itemized response to the editorial requests. The questions are italicized followed by our answers (Response) in which the red parts are the corresponding changes made in the manuscript. The corresponding changes in the manuscript and supplementary data are also highlighted by red color.

1). Additionally, please remember to include the NCBI Geo accession number and reviewer token in the revised manuscript.

Response: Thank you for the reminder. We have deposited the raw sequencing data in the public repository -- NCBI Gene Expression Omnibus (GEO). The sequencing dataset are available through the NCBI Gene Expression Omnibus (GEO) under accession number GSE151202:

<https://www.ncbi.nlm.nih.gov/geo/query/acc.cgi?acc=GSE151202>

The access token for reviewers is qzpcpgckndontep.

We have also revised the statement highlighted by red color in the manuscript (**Revised manuscript, Page 20, Data availability section**).

2). Please complete or update the following checklist(s) to verify compliance with our research ethics and data reporting standards. Address all points on the checklist, revising your manuscript in response to the points if needed. Each form should be uploaded as a Related Manuscript file at the time of resubmission: Editorial policy checklist; Reporting summary.

Response: Thank you for reminding. We have completed the Editorial policy checklist form and updated the Reporting summary form. They are uploaded as a Related Manuscript file. Please check them in the manuscript submission system.

3). Please confirm in your cover letter whether your study is compliant with the "Guidance of the Ministry of Science and Technology (MOST) for the Review and Approval of Human Genetic Resources", which requires formal approval for the export of human genetic material or data from China.

Response: Thanks for your kind reminder. Our study has been compliant with "Guidance of the Ministry of Science and Technology (MOST) for the Review and Approval of Human Genetic Resources" in China.

4). All Nature Communications manuscripts must include a "Data Availability" section after the Methods section but before the References. All novel microarray, DNA sequencing, RNA-seq and other sequencing datasets, or proteomic datasets must be deposited in a publicly accessible database, and accession codes, reviewer access and associated hyperlinks must be provided in the "Data Availability" section.

Response: Thanks for pointing out this. We have deposited the raw sequencing data

in the public repository -- NCBI Gene Expression Omnibus (GEO) and added the statement in the Data availability (**Revised manuscript, Page 20**).

5). *Please also include a “Code Availability” section after the “Data Availability” section.*

Response: Thank you for reminding. We have added the “Code availability” section after the “Data Availability” section (**Revised manuscript, Page 20**).

6). *To maximise the reproducibility of research data, we strongly encourage you to provide a file containing the raw data underlying the following types of display items. The data should be provided in a single Excel file with data for each figure/table in a separate sheet, or in multiple labelled files within a zipped folder. Name this file or folder ‘Source Data’, and include a brief description in your cover letter. The “Data Availability” section should also include the statement “Source data are provided with this paper.” Please replace your bar graphs with plots that feature information about the distribution of the underlying data. All data points should be shown for plots with a sample size less than 10. For larger sample sizes, please consider box-and-whisker or violin plots as alternatives. Measures of centrality, dispersion and/or error bars should be plotted and described in the figure legend.*

Response: Thanks for your kind reminder. Our sample size is less than 10, thus we have replaced the bar graphs with scatter plots in Supplementary Fig. 2e to feature information about the distribution of the underlying data. The data have been provided as a single Excel file for each figure, please check it in the Source data. We have included the statement “Source data underlying Figs. 2e are provided as a Source Data file” in the Data availability section (**Revised manuscript, Page 20**).

Also, we defined that n number refers to the number of patients in the Methods section - Histology analysis and Flow cytometry (**Revised manuscript, Page 20**).

Besides, according to the policy about Data availability, we added the sample size in the supplementary figure legend section -Supplementary Fig.2 (**Revised supplementary figure legends, Page 20**).

Meanwhile, we added the statement into Methods section to describe the process of bulk RNA-seq library preparation and sequencing (**Revised manuscript, Page 18**).

7). *Nature Communications is committed to improving transparency in authorship. As part of our efforts in this direction, we are now requesting that all authors identified as ‘corresponding author’ create and link their Open Researcher and Contributor Identifier (ORCID) with their account on the Manuscript Tracking System prior to acceptance. ORCID helps the scientific community achieve unambiguous attribution of all scholarly contributions. You can create and link your ORCID from the home page of the Manuscript Tracking System by clicking on ‘Modify my Springer Nature account’ and following these instructions. Please also inform all co-authors that they can add their ORCIDs to their accounts and that they must do so prior to acceptance.*

Response: Thank you for the reminder. The corresponding authors have linked their ORCID with their account on the Manuscript Tracking System. All other co-authors have been informed that they could add their ORCIDs to their accounts prior to acceptance.

3. Point-by-point responses to reviewers

Reviewer #1

Li et al. in their manuscript entitled “Single-cell transcriptome profiling of the vaginal wall in women with severe anterior vaginal prolapse” used single-cell RNA-seq to construct a transcriptome atlas of 81,026 individual AVP cells and control samples and identified 11 cell types in 16 women with stage III-IV anterior vaginal prolapse compared to 5 controls. This is the first study to examine single cell gene expression changes. They found dysregulation of the extracellular matrix and immune-complex reactions, including associated transcription factors and cell-cell communication. This paper represents a resource for enhanced understanding of cellular heterogeneity in women with pelvic organ prolapse. I have listed a number of comments on the supplementary tables as this is the data of the study and allows others to reproduce the results generated if they so wish.

comments:

1). Abstract – lines 30-31 are essentially repeated in lines 32-33.

Response: We thank the reviewer for pointing out this. We have revised the statements in the revised manuscript as follows (**Revised manuscript Page 1, Line 6-9**): “We revealed aberrant gene expression in diverse cell types in AVP. Intriguingly, unprecedented extracellular matrix (ECM) dysregulation and immune reactions involvement with prolapse were identified and enriched in both non-immune and immune cell types.”

2). Line 162 states that fibroblasts altered in POP tissue, although the proportion is the same as in the control tissue. Based on the previous sentences it is unclear why fibroblasts are altered in the prolapse samples. Some comments on the immunostaining and fluorescence-activated cell sorting differences might account for the alterations in the fibroblasts.

Response: Thanks for this thoughtful comment. We apologize for presenting this part unclearly. According to our IHC staining, the proportions of fibroblasts were similar between POP and control samples. Whereas, the proportion of epithelial cells and smooth muscle cells in POP samples was decreased. The statement in Line 162 was wrongly written, and it should be “epithelial cells and smooth muscle cells are altered in the prolapsed vagina”. We have corrected this in the revised manuscript (**Revised manuscript Page 6, Line 11**).

3). *The supplementary tables are very important to make sure that they are double checked for errors. There are several tables (e.g., Supplementary Tables 4 and 5) where gene names are cut-off because they extend beyond the allotted space. It would be most helpful to use a spreadsheet program such as Excel to display the tables so that they can be sorted and full names displayed rather than using a PDF format.*

Response: Thank for your valuable comments and suggestions. We are sorry for choosing the inappropriate format in the submission system when uploading supplemental files. We have re-uploaded all supplementary tables using Excel format.

4). *In the clinical characteristics supplementary table (Table 1), 21 POP cases are listed and 12 controls are listed, but only 16 cases are described in the text and 5 controls.*

Response: Thanks for pointing out this. Sorry for the confused description. In the scRNA seq analysis, we included 16 cases and 5 controls. The others listed in the supplementary table 1 were used for IHC or FCAS validation. We have added the descriptions and re-uploaded this supplementary table.

5). *While BMI is a risk factor for POP, the major risk factors for POP are not listed for the individuals who are part of your study including vaginal birth, family history of POP, and birth weight of infants. Please include this additional information if it is available.*

Response: Thanks for this thoughtful comment. We have newly included this additional information in the **Revised Supplementary Table 1**.

6). *The supplementary bulk DEG table (Table 4) contains Gene IDs rather than Gene names. It would be much more useful if it contained Gene names in addition to Gene IDs.*

Response: Thanks for this thoughtful comment. We have newly added Gene IDs in the bulk DEG table in the **Revised Supplementary Table 4**.

7). *The supplementary Table 5 includes gene enrichment descriptors that were not included in the manuscript (Figure 2) such as skin development that had the highest p-value. While skin development is not typically thought to be associated with POP, all terms should be included to be unbiased to the generated results. Also, note that some of the values in this supplementary table are cut-off and cannot be read in full. Again, a spreadsheet format would solve this problem.*

Response: Thanks for this thoughtful comment. We have revised Figure 2b which include the raw top 5 terms in the supplementary Table 5 (**Revised Fig.2b**). We have also revised and re-uploaded all supplementary tables and make sure they are fully

exhibited.

8). *Supplementary Table 6 contains the DEGs for the 11 clusters. The clusters are not provided and note that one of the genes listed is 3-Mar, likely an inadvertent spreadsheet change.*

Response: Sorry for the confused demonstration. The incomplete display may be due to that the submitting system transfer Excel to PDF format when we uploaded the file in the wrong file type. We have re-uploaded the Excel tables.

Reviewer #2

In this paper Li et al use single cell transcriptomics to investigate severe anterior vaginal prolapse. The authors contrast data between AVP cases and tissue from matched healthy controls. This allows them to deconvolve known molecular and histological phenotypes into cell type specific changes in distribution and molecular activity. All analysis is sound and on the refreshingly simpler side compared to many other papers making use of scRNA-seq technology. The paper is straightforward and clearly contributes to knowledge about AVP. It sets up a next step of making use of the identified cellular and molecular changes to study causes of AVP. Only some minor issues about reporting need to be addressed. Minor issues:

1). *In figures it is not clear which abbreviation corresponds to which cell type. A legend would be helpful, either in the figure, in the text, or in the figure legend.*

Response: Thanks for pointing out this. To clearly describe the abbreviation corresponding to the cell type, we have newly added the full name and abbreviation of cell types in the **figure legend**.

2). *When using scRNA-seq to study differences in proportions of cell types the authors should report how large the differences are, and a certainty measurement such as a P-value. In the first instance differential cell type proportions are studied, proportions are quantified and validated using IHC and FACS. But such measurements are not done in the analysis of cell subtypes. Quantifications of differences of proportions are needed for Fig.S7, Fig.S8, and Fig.S9. It would also be good to add this to Fig.S2 even though quantification is performed using other measurements.*

Response: Thanks for this thoughtful comment. For the cell proportion investigation, we employed Ma's method that published in *Cell* as reference¹. By calculating the Log₂FC between two groups ($|\text{Log}_2\text{FC}| > 0.5$) to identify the cell types alteration along different conditions, the value of $|\text{Log}_2\text{FC}|$ represents the difference. We have revised **figure s2**, **figure s7**, **figure s8** and **figure s9**. Furthermore, we also added the additional file with raw value of cell proportion for reference in **Supplementary Table 7**.

Moreover, we have designed immunofluorescence staining to characterize subtypes in fibroblasts, smooth muscle cells and macrophages. We chose the subtype-specific marker genes according to their specific expression and their cellular localization. Three control samples and four POP samples were used for each subtype. As the results shown, we successfully validated the existence of four main subtypes of fibroblasts ($CTSB^+$, $HMGAI^+$, $CD55^+$, and $C7^+$ fibroblasts) (**Rebuttal Fig. 1**) and two main subtypes of smooth muscle cells ($TM4SFI^+$ and $CYCS^+$ smooth muscle cells) in control and POP samples, respectively (**Rebuttal Fig. 2**). Meanwhile, we applied conventional surface markers and characterized subtype 1 of macrophages (i.e. phagocytic macrophages, CD206) and subtype 2 of macrophages (i.e. inflammatory macrophages, CD86) (**Rebuttal Fig. 3**). On the other hand, we quantified the positive staining cells (**Rebuttal Fig. 4**). Whereas the expression of subtype markers showed different trends among samples and on condition of smaller numbers of patient samples, the quantification results have no significant difference. Thus, we suggest the quantification of subtypes could only be as a reference.

Rebuttal Fig. 1. Characterization of subtypes in fibroblasts. a. UMAP plot showing the expression of cell subtype specific markers. b. Immunofluorescence co-staining confirmation of the existence of $CTSB^+$, $HMGAI^+$, $CD55^+$, or $C7^+$ subtype of fibroblasts. Fibroblasts are marked using Vimentin. Immunostaining images shown in this figure are representative of the marker staining patterns observed in control samples and POP samples. Scale bar represents 50 μm .

Rebuttal Fig. 2. Characterization of subtypes in smooth muscle cells. a, c. UMAP plot showing the expression of cell subtype specific marker. b, d. Immunofluorescence co-staining confirmation of the existence of *TM4SF1*⁺, or *CYCS*⁺ subtype of smooth muscle cells. Smooth muscle cells are marked using α -SMA. Immunostaining images shown in this figure are representative of the marker staining patterns observed in control samples and POP samples. Scale bar represents 50 μ m.

Rebuttal Fig. 3. Characterization of subtypes in macrophages by conventional surface markers. a. Immunofluorescence staining of CD206 for phagocytic macrophages. b. Immunofluorescence staining of CD86 for inflammatory macrophages. Immunostaining images shown in this figure are representative of the marker staining patterns observed in control samples and POP samples. Scale bar represents 50 μ m.

Rebuttal Fig. 4. Quantification of positive staining of subtypes in fibroblasts, smooth muscle

cells and macrophages.

Reviewer #3

This is a well-written paper on scRNA-seq analysis of anterior vaginal wall tissues from patients with prolapse compared to controls. My comments are listed below.

1). Much of the findings from this analysis is consistent with what is already published, for example, alterations in ECM, smooth muscle cell content, and immune/inflammatory pathways. The process of tissue keratinization in prolapse is also well documented since the tissues undergo keratinization from exposure to the outside environment and downward pressure. While this increases confidence that the scRNA-seq is yielding reliable data, it not novel.

Response: Thanks for point this. We agreed with the reviewer's concern about some findings that are consistent with other studies such as ECM, smooth muscle cells content. However, we found that the alterations in ECM were widely involved in most of cell types, which was not realized in other studies. More importantly, the expression patterns of known ECM genes related to POP differed in each cell type, even differed in each subtype. This finding might provide explanations why the reported alterations of ECM such as collagen, elastin in other studies were contradiction. This also inspired that it's important and valuable to dig out the mechanism in the single-cell level. To the best of our knowledge, our study is the first study to examine single cell gene expression changes in POP.

Besides, our study also revealed dysregulated cell-cell communication patterns and identified abnormal transcription factors in AVP, which provide a valuable resource for deciphering the cellular heterogeneity and the molecular mechanisms underlying severe AVP.

2). What is novel is the finding of different sub-groups of fibroblasts and smooth muscle cells. However, this finding is not well developed in the manuscript. Figure 5 is confusing in that it is difficult to see how POP and controls differ in the distribution of these subgroups. Further characterization and verification of these subgroups would be of interest to the scientific community. scRNA-seq is a good methodology to investigate the rare sub-groups. These data may reveal interactive mechanisms that have not been considered.

Response: Thanks for the valuable comments and suggestions. We apologies for presenting this part unclearly. Until now, the extent of cell heterogeneity in fibroblasts, smooth muscle cells and macrophages is unexplored. To better understand the difference in the sub-groups, we visualized the changes through cell distribution and cell proportion in each subtype between control and POP samples. Meanwhile, we also investigated the gene expression on transcription factors and ligand-receptor pairs of each subtypes compared with control samples. The important transcription factors and ligand-receptor pairs that were identified in cell clusters exhibited different

expression patterns in each subtype. In addition, the corresponding statements have been revised in the manuscript result section “**Identification of cell subtypes and cellular alterations in POP samples**” (Revised manuscript Page 11, Paragraph 2) and **Figure 5** has been revised as follows:

Revised Fig. 5. Subclustering of major cell types in the vaginal wall reveals cellular heterogeneity. a. UMAP plot showing the distribution of 7 distinct fibroblasts subtypes in control and POP samples. b. Bar plots showing the cell proportion of 7 fibroblast subtypes in control and POP samples. c. Relative expression of representative DEGs, TFs and ligand-receptor pairs among 7 subtypes in POP samples than that in control samples. d. Subclustering of smooth muscle cells further identified 4 distinct subtypes. e. Bar plots showing the cell proportion of 4 smooth muscle cells subtypes in control and POP samples. f. Relative expression of representative DEGs, TFs and ligand-receptor pairs among 4 subtypes b in POP samples than that in control samples. g. Subclustering of macrophages further identified 5 distinct subtypes. h. Bar plots showing the cell proportion of 5 macrophages subtypes in control and POP samples. i. Relative expression of representative DEGs, TFs and ligand-receptor pairs among 5 subtypes in POP samples than that in control samples.

Besides, we have performed immunofluorescence staining to characterize the main subtypes in fibroblasts, smooth muscle cells and macrophages. We chose several subtype-specific marker genes according to their expression and their cellular localization (**Rebuttal Fig. 1a and Rebuttal Fig. 2a, c**). As the results shown, we validated four main subtypes of fibroblasts ($CTSB^+$, $HMGAI^+$, $CD55^+$, and $C7^+$ fibroblasts) (**Rebuttal Fig. 1b**) and two main subtypes of smooth muscle cells

(*TM4SF1*⁺ and *CYCS*⁺ smooth muscle cells) (**Rebuttal Fig. 2b, d**). Meanwhile, we applied conventional surface markers and characterized subtype 1 of macrophages (i.e. phagocytic macrophages, CD206) and subtype 2 of macrophages (i.e. inflammatory macrophages, CD86) (**Rebuttal Fig. 3**).

2). Please provide a table with patient characteristics and statistical comparisons of these differences in age, BMI, and parity can confound the data.

Response: Thanks for your valuable comments. We have compared the differences of cell distribution, proportion and DEGs of samples in different age, BMI and parity. We divided POP patients into different groups according to the following rules: Age groups (<65 age, ≥65 age); BMI groups (18.5-24.9, ≥25); Parity groups (=1, >1). Then each group was compared with control sample, respectively. We found that cell subtypes distribution and cell proportion in the <65 age group and ≥65 age group have no significant difference (**Rebuttal Fig. 5-7**). The same results were observed in two BMI groups and two Parity groups separately (**Rebuttal Fig. 5-7**). As an example, in the aspect of DEGs, we selected top-20 upregulated genes and top-20 downregulated genes from the <65 age group among each subtype based on the Log₂FC. The heatmap indicated that these genes showed the similar expression trends in the ≥65 age group (**Rebuttal Fig. 8**). GO enrichment results of two Age groups also showed the same trend, in which the upregulated genes were involved in the extracellular matrix organization in most cases. Nevertheless, we found that the expression of some genes has slightly difference. For example, although *THBS4*, *LYVE1* and *MMP11* were upregulated in both two age groups and *HAS2* was downregulated in both two age groups, the relative expression level was different (**Rebuttal Table 2-37**). The same studying strategies were performed in two BMI groups and two parity groups, respectively. The same results were got in two BMI groups and two parity groups. The same expression trends but the different expression level in a certain gene are partly in consistent with the influence of age, BMI and parity which are associated risk factors of POP.

Rebuttal Fig. 5. Distribution of fibroblasts subtypes across different conditions. a-c. UMAP projection of 44,965 cells under different age range, BMI and parity. d. UMAP plot showing fibroblasts subtypes in control (n=5) samples. e. Dot plot depicting the change of subtype cells proportion in POP samples across different conditions compared with control sample (red, increased; blue, decreased; grey, no significant change).

Rebuttal Fig. 6. Distribution of smooth muscle cells subtypes across different conditions. a-c. UMAP projection of 14,562 cells under different age range, BMI and parity. d. UMAP plot showing fibroblasts subtypes in control (n=5) samples. e. Dot plot depicting the change of subtype cells proportion in POP samples across different conditions compared with control sample (red, increased; blue, decreased; grey, no significant change).

Rebuttal Fig. 7. Distribution of macrophages subtypes across different conditions. a-c. UMAP projection of 6,082 cells under different age range, BMI and parity. d. UMAP plot showing fibroblasts subtypes in control (n=5) samples. e. Dot plot depicting the change of subtype cells proportion in POP samples across different conditions compared with control sample (red, increased; blue, decreased; grey, no significant change).

Rebuttal Fig. 8. Differential expression genes in different cell subtypes across different conditions. a-c. Heat map showing the normalized expression levels of top 20 upregulated genes and top 20 downregulated gene (see the methods) in different conditions compared with control samples. Red color indicates upregulated, while blue color indicates down regulation ($p < 0.05$, $|\text{Log}_2\text{FC}| > 0.5$).

Reference

1. Ma, S. *et al.* Caloric restriction reprograms the single-cell transcriptional landscape of *rattus norvegicus* aging. *Cell* **180**, 1-18, doi:10.1016/j.cell.2020.02.008 (2020).

Reviewers' Comments:

Reviewer #1:

Remarks to the Author:

The authors have addressed my concerns.

Reviewer #2:

Remarks to the Author:

In this review Li et al have addressed most comments raised by the reviewers.

Only a slight issue remains. The authors were asked to quantify the certainty of the control vs AVP patient differences in cell type proportions, but did not do this. At the very least, the authors should change the cell type proportion panel in Figure 5 to have one column per patient, so researchers in the future can quantify the certainty of the difference in cell type proportions.

The authors have now made their scRNA-seq data available, but it does not include the authors cell type labels. Cell type assignment is also not covered in the R analysis script made available by the authors.

Reviewer #3:

None

Revision Summary

We want to thank the editors for their thoughtful advice and the reviewers for their critical and insightful comments. We have now revised the figures according to the reviewer's comment. We also revised the manuscript to comply with the editorial requests. Point-by-point responses to the editors and the reviewers' comments are included. The questions are italicized followed by our answers. The corresponding changes in the manuscript and supplementary data are highlighted by red color.

1. Point-by-point responses to the editor

1) At the same time we ask that you edit your manuscript to comply with our policies and formatting requirements and to maximise the accessibility and therefore the impact of your work. Please see the attached document(s), listing a number of points that must be addressed.

Response: Thank you for thoughtful work. We have thoroughly revised the manuscript and figures according to the editorial requests. Please check the detailed response in the revised author checklist, reporting summary, updated editorial policy checklist, revised manuscript, supporting information, and supplementary files.

2. Point-by-point responses to reviewers

Reviewer #2

In this review Li et al have addressed most comments raised by the reviewers.

comments:

1) Only a slight issue remains. The authors were asked to quantify the certainty of the control vs AVP patient differences in cell type proportions, but did not do this. At the very least, the authors should change the cell type proportion panel in Figure 5 to have one column per patient, so researchers in the future can quantify the certainty of the difference in cell type proportions.

Response: Thank you for the advice. We have added the new graph to demonstrate the cell type proportion per patient in the **Revised Figure 5**.

2) *The authors have now made their scRNA-seq data available, but it does not include the authors cell type labels. Cell type assignment is also not covered in the R analysis script made available by the authors.*

Response: We thank the reviewer for pointing out this. We have provided Supplementary data 2 for cell type assignment and updated the R analysis script with cell type assignment in GitHub.